# The Iron(ic) Melting Pot:
# Reviewing Human Evaluation in Humour, Irony and Sarcasm Generation

**Tyler Loakman[1], Aaron Maladry[2] and Chenghua Lin[1, 3] [*]**

[1]Department of Computer Science, The University of Sheffield, UK
[2]LT[3], Ghent University, Belgium
[3]Department of Computer Science, The University of Manchester, UK

tcloakman1@sheffield.ac.uk
aaron.maladry@ugent.be
chenghua.lin@manchester.ac.uk

## Abstract

Human evaluation is often considered to be the gold standard method of evaluating a Natural Language Generation system. However, whilst its importance is accepted by the community at large, the quality of its execution is often brought into question. In this position paper, we argue that the generation of more esoteric forms of language - humour, irony and sarcasm - constitutes a subdomain where the characteristics of selected evaluator panels are of utmost importance, and every effort should be made to report demographic characteristics wherever possible, in the interest of transparency and replicability. We support these claims with an overview of each language form and an analysis of examples in terms of how their interpretation is affected by different participant variables. We additionally perform a critical survey of recent works in NLG to assess how well evaluation procedures are reported in this subdomain, and note a severe lack of open reporting of evaluator demographic information, and a significant reliance on crowdsourcing platforms for recruitment.

## 1 Introduction

The aim of a Natural Language Generation (NLG) system is to generate coherent and well-formed text of a particular type, usually given an input such as a prompt (Raffel et al., 2020; Wang et al., 2022), outline (Goldfarb-Tarrant et al., 2020; Tang et al., 2022), topic (Van de Cruys, 2020), or data (Kale and Rastogi, 2020; Chen et al., 2020). As a result, NLG contains many sub-domains such as summarisation (Goldsack et al., 2022, 2023), data-to-text generation (Kale and Rastogi, 2020; Chen et al., 2020), dialogue generation (Tang et al., 2023), story generation (Goldfarb-Tarrant et al., 2020; Tang et al., 2022), and humour generation (Sun et al., 2022b; Loakman et al., 2023). Whilst it may be possible for any competent speaker of the target language to sufficiently understand generated outputs, many language forms may be considered

more esoteric, such as humour, irony, and sarcasm - where interpretations of such language are highly intertwined with more broad demographic characteristics, as well as individual and idiosyncratic factors.

Human evaluation is viewed as the foremost important form of evaluation for any NLG system (Howcroft et al., 2020), owing largely to its direct capture of the opinions of the (usually) human end-user.[1] This is made even more prominent when considering the low concurrent validity that many easily deployed automatic evaluation metrics have with human judgements (Reiter, 2018; Alva-Manchego et al., 2021; Zhao et al., 2023a). Whilst such evaluation procedures are considered preferable over automatic metrics (particularly in the cases where no reliable automatic metrics are available), this does not mean that they are infallible. Numerous recent works have outlined the inconsistency with which human evaluation procedures are executed, and increasing attention is being brought to the knock-on effects that such oversights may cause (Zajko, 2021).

In this paper, we present an overview of 3 particular types of language: humour, irony and sarcasm - and demonstrate how different evaluator demographics[2] may affect the interpretation of such language, and therefore its evaluation in relevant systems. Furthermore, we present a critical survey and assessment of 17 NLG research papers from between 2018 and 2023 from the ACL Anthology that concern the generation of these types of language, and analyse behaviour in regard to the level of transparency used when reporting details of the human evaluation procedure.

## 2 Related Works

In recent years, the execution of human evaluation in NLG has become an even more significant topic, with myriad top-tier venues hosting dedicated tracks and workshops specifically focused on the evaluation

---

[*] Corresponding author

[1]Sometimes NLG outputs are used as inputs to further NLG models (Zhang et al., 2022).

[2]In what we describe as a "cultural melting pot" in the title.

of evaluation *itself*, such as the HumEval workshops at EACL 2021, ACL 2022 and RANLP 2023.[3] As a product of such venues, multiple recent works have been proposed that aim to highlight common inadequacies in multiple aspects of human evaluation, including the design of assessment criteria, the recruitment and training of participants, the use of biased response elicitation techniques (Parmar et al., 2023), and the transparency of reporting these processes.

Howcroft et al. (2020) assess a 20 year time-span of work within NLG (165 papers between 2000-2019) in regard to how evaluation procedures are performed, finding that there is substantial variation in how evaluation criteria are named and portrayed. Resulting from this, they produce normalised mappings between different criteria descriptions and their underlying evaluation focus and call for human evaluation criteria to be standardised, in addition to the creation of evaluation sheets outlining best practices for response elicitation from evaluators. Amidei et al. (2019) also assess NLG research in terms of response elicitation methods with a particular focus on the use of Likert scales and common deviations from best-practice within 135 papers from a 10 year period (e.g. whether or not to view them as ordinal or interval in nature, and consequently how to perform statistical significance testing). Further issues have also been identified by Schoch et al. (2020) with the framing of evaluation questions and participant recruitment methods, including the presence of demand characteristics in evaluators being more common on crowdsourcing platforms as participants vie for limited positions in HITs and wish to give researchers the results they believe they desire (under the belief it will lead to more future work).[4] Furthermore, these issues are exacerbated by cases wherein researchers reject annotations outright due to not falling in line with their expectations, or pay inadequately for services on crowdsourcing platforms (Huynh et al., 2021; Shmueli et al., 2021). However, whilst automatic metrics may have many drawbacks for evaluating NLG, human evaluation has been suggested to be imperfect by its very nature, with human judgements being inconsistent. For example, Clark et al. (2021) find that human evaluators range significantly in their preconceptions of the quality of text NLG systems can produce, resulting in evaluators having biased perspectives when assessing text quality (a bias

that is only somewhat overcome with the provision of training materials to standardise criteria definitions among evaluators). In order to partially mitigate these evaluation pitfalls, van der Lee et al. (2019) suggest a range of best practices, including the reporting of confidence intervals for results, using multiple ratings of each criteria, and using randomisation and counterbalancing when assigning evaluators to conditions (e.g. human- vs model- generated text). Additionally, due to many of the problems that we outline in this work, the domain is currently starting to demonstrate the progression of a paradigm shift towards understanding the role of human disagreement as a valuable feature, rather than noise, which is also resulting in the rise of perspectivist approaches (Pavlick and Kwiatkowski, 2019; Meaney, 2020; Plank, 2022; Uma et al., 2022). Additionally, whilst the use of such language can lead to different responses from different people, this effect is compounded in human-computer interactions, where the communicative divide is even more pronounced (Shani et al., 2022; Shapira et al., 2023).

## 3 Introducing Humour, Sarcasm, and Irony

In the following section we discuss three primary examples of language types where interpretations and opinions are highly affected by a range of nuanced participant variables. For each type, we outline the nature of this language alongside a range of examples, and reflect upon how interpretation of these examples may differ across demographics.

### 3.1 Humour

As the first language form to discuss, humour is also one of the most complex. The general consensus within cognitive and linguistic theories of humour is that an essential element for something to be found humorous is the existence of some degree of incongruity between expectations and reality (Raskin, 1985). Consequently, the experience of humour arises from the resolution of these incongruities within someone's mind. This, in turn, is how puns may elicit humour, when the initial interpretation of a text primes the interpretation of one semantic sense of a word, before that expectation is violated when the text is viewed as a whole. For example, in "the greyhound stopped to get a hare cut", the interpretation of "hare" as in the rabbit-like animal is primed by the mention of the "greyhound" which chases hares, whilst "hare cut" forces the reader to reinterpret "hare" as the homophone "hair", which resolves the incongruity from the non-collocational phrase "hare cut" (He et al., 2019).

---

[3] https://humeval.github.io

[4] Demand characteristics are cues that may allow a participant to infer the nature and desired outcome of a study, often resulting in the participant then altering their behaviour to align with the perceived expectation, thereby invalidating results.

| Type | Example |
|---|---|
| Humour | 1. Q: "Why would Trump change a lightbulb?" A: "He was told Obama installed it." |
| | 2. Q: "How does a man change a lightbulb?" A: "He holds it and waits for the world to revolve around him." |
| Non-humour | 3. "Colourless green ideas sleep furiously." |

Table 1: Examples of humorous jokes with different types of assumed world knowledge.

Whilst puns are the most frequently studied form of humour in the area of NLG (He et al., 2019; Mittal et al., 2022; Tian et al., 2022), primarily due to their simple exploitation of phonetics and word senses, humour more generally can arise from more complex incongruities which are more likely to be resolved by individuals with specific assumed world knowledge. Table 1 presents examples of such humour. In Example 1, the perception of humour is dependent on factors including an individual's knowledge of politics and their ideological leaning, as well as region.[5] The reader is therefore assumed to have external background knowledge of US politics in order to infer that the joke concerns Donald Trump's record of reversing actions made by his predecessor, Barack Obama. Additionally, due to the joke using Trump as the target and in effect calling him petty, whether or not this joke is seen as humorous is also dependent on one's political leanings. This is due to Republicans, the political party that Trump represented, being less likely to find comments mocking him to be humorous. In Example 2, the joke is based around the shared knowledge of individuals in a patriarchal society which has, of course, traditionally benefited men. As a result, the joke uses men as the target, and comments on a presumed level of arrogance. In addition, this example requires the reader to interpret both literal and metaphorical meanings of something "revolving around" something else. Consequently, factors that could affect the perception of humour here are one's familiarity with a patriarchal society and culture (as it can be imagined that such assumptions about male arrogance are not shared by members of a matriarchal culture). Additionally, due to the joke targeting men, there is the possibility for some men to be offended at the assumptions that it is making, and it is known that gender generally affects the perception of certain types of jokes such as these (Lawless et al., 2020). Furthermore, a large body of work has investigated the different tendencies of particular genders towards different patterns of language use,

which in some cases may also affect the perception of different language types (Coates, 2016; Rabinovich et al., 2020). Finally, Example 3 (Chomsky, 1957) is an instance of an incongruous sentence which is semantically nonsensical (yet grammatically valid), where this incongruity does not give rise to humour (in the general case). In addition to the examples outlined and analysed here, numerous works have also ascertained a link between different demographic variables and humour perception. For example, Jiang et al. (2019) explore the literature on how individuals in the Western and Eastern worlds view humour as suitable under different conditions, and differ also in the exact forms of humour they opt to produce.

## 3.2 Irony

Although the exact definition of irony still remains a topic of discussion, NLP research has reached the general consensus that verbal irony is a form of figurative language where the producer of the language intends to convey the opposite meaning of the literal interpretation (Grice, 1978; Burgers, 2010; Camp, 2012). Consequently, similar to within the humour domain, an ironic statement violates the true beliefs of the producer and the expectations of the audience. It is for this reason that many ironic (and sarcastic) statements may also be considered humorous - a well known feature of British comedy programming (Gervais, 2011). This means that correct understanding of irony presupposes true or accepted knowledge about whatever thing, person or situation is being referred to. Such presupposed knowledge can assume multiple forms. A range of ironic texts are presented in Table 2. Irony can, for example, be factual, where it is generated by making a factually incorrect statement such as in Example 1 where a human alludes to having wings. Alternatively, it can also rely on social conventions, habits or shared knowledge within members of a community. For instance, in order to understand the irony in Example 2, the receiver is expected to know that most people would not genuinely applaud the actions of someone they call a dictator. Whilst some knowledge and social

---

[5]In this case, US politics is quite well known world-over, but a similar joke for a different government could be imagined, where universal knowledge would be far less likely.

| Type | Example |
|------|---------|
| Irony | 1. "Just found out about the train strike today, guess I'll have to use my wings and fly there myself." |
| | 2."It's been a great week for dictators. Congrats North Korea and Cuba." |
| | 3. "Never knew something could be 80% halal... I've learned something new I guess." |
| | 4. "A local fire station has burnt down." |
| Non-irony | 5. "I went to the hospital yesterday. It was a big building." |

Table 2: Examples of different forms of verbal irony (excluding sarcasm).

conventions are ubiquitous, many ironic expressions also assume specific (in-group) knowledge that is often only shared within a particular social group. For example, within the statement seen in Example 3, the receiver of the message is required to understand that "halal" is a most often used as a binary concept, and therefore something being a "percentage" halal is infeasible, thus requiring additional background knowledge about Islam and its conventions. A single individual can, of course, be a member of several different social groups that share common ideals and knowledge, such as political conviction, free-time activities, or music taste, without even having to consider major cultural or geographical divergences. This simple fact, that irony strongly relies on common(sense) knowledge, makes it an inherently subjective phenomenon, both for detection and the evaluation of generative models. In addition, ironic statements can also rely on situational or contextual knowledge, in which case it is then classified as "situational irony" as opposed to "verbal irony". An example of situational irony is demonstrated in Example 4, where a fire station burning down is unexpected due to containing multiple resources explicitly for the extinguishing of such fires. According to Wallace et al. (2014), contextual information may even be essential for most humans to recognise most forms of irony. Finally, Example 5 presents a non-ironic statement, where expectations about concepts are not violated, as a hospital is usually assumed to be a large building. Recent works have taken into account the plethora of variables affecting the interpretation of irony, with Frenda et al. (2023) developing a corpus of irony from a perspectivist approach, finding that irony perceptions vary with generation and nationality (even within speakers of the same language).

### 3.3 Sarcasm

Traditional definitions usually consider sarcasm to be a sub-category of irony. In addition to having a strong negative connotation and aggressive tone, this form of verbal irony is intended to ridicule someone or something (Filik et al., 2019). Table 3 presents a selection of sarcastic texts. The clearly aggressive intent inherent in sarcasm is presented in Examples 1 and 2. In Example 1, the intention is to express that someone with seemingly no special qualities is able to have a sense of pride, whilst in Example 2, the choice of the word "marinate" implies that the target of the comment is wearing too much perfume, similar to how meat may marinate in a glaze for many hours before being ready. Examples 3 and 4 present non-sarcastic utterances, with Example 3 being a blatant insult, and Example 4 demonstrating the expected gratitude someone would have if they had been bought a supercar (assuming no deeper contextualised meaning). As sarcasm is considered to be a subcategory of irony, research into irony detection or generation may often include sarcasm, and treats them as one and the same (Jijkoun and Hofmann, 2009; Filatova, 2012; Van Hee, 2017; Maladry et al., 2022). Moreover, studies by Gibbs (1986) and Bryant and Fox Tree (2002) indicate that there is an ongoing semantic shift, where verbal irony is perceived more often as sarcasm in popular use. Recent papers on sarcasm generation (Mishra et al., 2019; Oprea et al., 2021) do not systematically discuss their definition of sarcasm in relation to the concept of verbal irony. However, one has to assume here that sarcasm is synonymous with verbal irony, since it would be suboptimal for generative systems to create content that might ridicule people in a hurtful way. In short, both for pragmatic reasons as well as conceptual changes, it makes sense to consider irony and sarcasm to be the same thing. Thus, the same motivations that make irony subjective apply to sarcasm as well. On top of that, detection and generation tasks concerning sarcasm specifically also demand that evaluators are able to discern the aggressive tone and retrieve the intention of the author or speaker, which can vary significantly. For example, Phillips et al. (2015) find age to be a

| Type | Example |
|------|---------|
| Sarcasm | 1. "It's astonishing how someone so average and unremarkable can manage to feel good about themselves." |
| | 2. "Nice perfume. How long did you marinate in it?" |
| Non-sarcasm | 3. "You're an idiot." |
| | 4. "Thank you so much for buying me a Ferrari!" |

Table 3: Examples of sarcasm and non-sarcasm.

significant contributing factor in an individual's ability to correctly perceive intended sarcasm, with older individuals struggling more than younger. Furthermore, Rockwell and Theriot (2001) find gender differences in the patterns of sarcasm use to both be affected by the gender of the speaker, as well as the gender of the recipient interlocuter. Additionally, the use of sarcasm has been shown to vary alongside socio-cultural contexts, with Blasko et al. (2021) noting that sarcasm use is more prevalent in individualist countries with a lower power distance such as the U.S than collectivist countries with higher power distance such as China.[6] It logically follows, therefore, that these differences may also impact someone's ability to perceive intended sarcasm, and consequently how they would react to sarcastic utterances in different contexts.

## 4 Critical Survey

Following existing works that critically review human evaluation procedures in Natural Language Generation, we present a critical survey of works in the area of computational humour, irony, and sarcasm. We begin by reviewing the scope and range of our literature search for this review and the prerequisite criteria for incorporation. Following this, we motivate the need for increased care in this specific domain of language by briefly exploring the prevalence of research in the area, both in terms of generative works, as well as those on the detection of such language. As previously discussed, irony and sarcasm are highly related language forms, resulting in the common use of the terms interchangeably. However, to ensure that the intended definition is respected (the correctness of which cannot always be inferred with complete certainty) we report irony and sarcasm separately based on the titles of the papers.

### 4.1 Scope and Selection

The papers included in this survey pool were all taken from the ACL Anthology for the venues of *ACL, EMNLP, COLING, LREC, INLG, and CoNLL, between 2018 and 2023.[78] Initially, all papers concerning humour, irony and sarcasm were selected. This was performed by keyword matching titles on the following word stems: "hum-", "sarc-", "iron-", "fun-", "jok-" and "pun-".[910] The selected papers include main conference, Findings, and workshop submissions, in addition to one system demonstration, resulting in 259 papers. Submissions to shared-tasks were then excluded, reducing the pool to 135 papers.[11] The resulting papers were then checked over to ensure relevance, and categorised broadly into generation or detection (the latter of which may be binary, multi-class, or estimation). These criteria resulted in 135 remaining papers, of which 22 were focused on the generation of humour, sarcasm and/or irony, and 108 on detection (with the remaining 5 being classed as "other"). We then excluded 4 of the remaining 22 generation papers that were surveys or dataset papers where human evaluation was performed on non-computationally generated examples. This resulted in a final selection of 18 generation papers, all of which are listed in Appendix A.1.

### 4.2 Subject Prevalence

As seen in Figure 1, there has been a significant number of accepted papers on these topics within this 6 year scope of collection. In addition to these papers, this search also revealed 5 shared tasks within this period, including a sarcasm detection shared task

---

[6]"Power Distance" refers to the distribution of perceived power (political, financial etc.) across a society. A lower power distance means that members of a society are more "equal", for example, when comparing a blue collar worker to a politician.

[7]Where the relevant venues have occurred at the time of writing.

[8]INLG and CoNLL contributed 0 papers that met our criteria for inclusion, but were nevertheless surveyed.

[9]To capture the most common form of humour generation.

[10]In order to capture "humour", "sarcasm", "irony", "funny", "joke", and "puns", including all derivations and regional spelling.

[11]This is due to shared task submissions often attracting non-novel approaches, and all following a standardised evaluation procedure as determined by the task organisers. Their inclusion would therefore bias the results.

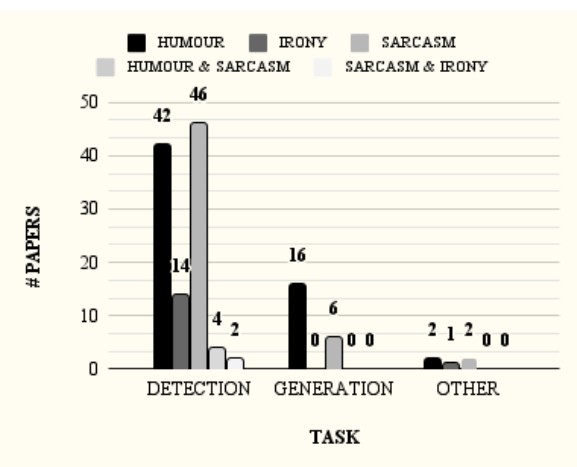

Figure 1: Breakdown of selected papers by NLP task and specific language type (based on the paper titles). These values exclude shared task papers, but include generation papers that are surveys or datasets, which were later removed. The "other" task category refers to papers primarily presenting content such as annotation schema (Pamulapati et al., 2020) and cognitive analyses (Zeng and Li, 2022), among other forms of research in this area we are not directly investigating.

at the Figurative Language Processing workshop at ACL 2020 (Ghosh et al., 2020), Task 7 of SemEval 2020 and 2021 on humour detection (Hossain et al., 2020; Faraj and Abdullah, 2021), a WANLP 2021 shared task on sarcasm detection (Abu Farha et al., 2021), and SemEval-2022 Task 6, also on sarcasm detection (Abu Farha et al., 2022).[12]

In addition to the works captured by our literature search and the aforementioned shared tasks, it is reasonable to think that the prevalence of recently released Large Language Models (LLMs) such as ChatGPT and GPT-4 would result in further increases in research into this field, now that models are widely available that are able to better handle the generation and explanation of complex topics such as humour. Not explicitly included in this survey, but nevertheless relevant, are also more general works on anthropomorphism in language-based AI agents (Abercrombie et al., 2021; Gros et al., 2022), in which the ability to detect, generate, and respond to humour, irony and sarcasm are paramount if anthropomorphism is deemed desirable (Oprea et al., 2022).

### 4.3 Reporting Transparency

Of the 135 remaining papers, we now move our attention to the 18 focused on generation specifically.

---

[12]As expected, all shared tasks were on detection rather than generation, due to the more simple nature of evaluation for a shared task environment.

All but 1 of the 18 papers found also contained human evaluation, and therefore the remaining 17 papers will form the basis of the following analysis. Following on from earlier sections which motivated the significance of participant variables in the interpretation of these language forms, Table 4 presents the percentage of surveyed papers that report on the following evaluator demographic information: language proficiency, location and nationality, age, gender, education level, and social class.

**Participant Demographics**   As Table 4 shows, less than half of the assessed works (7 of 17) reported on *any* demographic information regarding their selected human evaluator panels. Of those that were more transparent in regard to evaluator demographics, language proficiency and location (usually as a proxy for nationality) were the most common detail given. A single paper (Mishra et al., 2019) reported evaluator education.

Importantly, even in those papers which did outline some demographic information, this reporting was frequently crucially inadequate. For example, Oprea et al. (2022) provide the full participant information sheet that was presented, but reduce the demographic information to "We target everyone registered as living in ⟨country⟩ on the Prolific Academic platform.", where the valuable information of *which* country (or which countries) is redacted. Additionally, the one paper that reported education (Mishra et al., 2019) was only to the extent that the two evaluators "had a better understanding of the language and socio-cultural diversities" - a statement that may range from meaning native speakers of the target language in the desired culture, to people with doctorates in sociology. None of the surveyed papers outlined evaluator age, gender or social class.

**Recruitment**   As can be seen in Table 5, 12 of the 17 surveyed papers contained explicit mention of participant recruitment methods. Upon further investigation into these cases, we find that all 12 instances concern the use of crowdsourcing platforms, of which 9 are Amazon Mechanical Turk (AMT), 2 are Prolific Academic (Oprea et al., 2021, 2022), and one does not explicitly mention the platform (Tian et al., 2022). Of these 12 papers that report recruitment, compensation amounts are reported in less than half (5 papers), ranging from approximately

| Type | # Papers | Percentage |
|------|----------|------------|
| **Demographics?** | **7** | **<42%** |
| **Language Proficiency** | 4 | <24% |
| **Location/Nationality** | 4 | <24% |
| **Age** | 0 | 0% |
| **Gender** | 0 | 0% |
| **Education** | 1 | <6% |
| **Social Class** | 0 | 0% |

Table 4: Counts of papers containing reporting of demographic variables in the area of humour, sarcasm and irony generation. "Demographics" refers to the number/percentage of papers that report any demographic information.

9.23 USD[13] (Oprea et al., 2021, 2022) to 20 USD per hour (Tian et al., 2022; Mittal et al., 2022).

One reason that transparent reporting of recruitment methods is important is due to allowing inference of potential participant characteristics that may not be reported elsewhere. For instance, research in the social sciences has often been criticised for focusing primarily on **WEIRD** participants, **W**estern and **E**ducated individuals from **I**ndustrialised, **R**ich and **D**emocratic societies (Henrich et al., 2010). However, as access to language technology becomes more democratised, cheaper to access and easier to use (Simons et al., 2022; de Dios-Flores et al., 2022) the majority of end users of these technologies are not going to be well represented by this set of characteristics that participants often have. This tendency is further reflected by the overwhelming bias towards work on the English language, which leads to the demotion of other non-digitally accessible languages, and reinforces existing inequalities (Søgaard, 2022). In addition to these characteristics, there has also historically been a bias towards male participants in scientific research (Holdcroft, 2007; García-González et al., 2019), something that is also common in AI due to the field of computer science still being dominated by men overall (European Commission and Directorate-General for Research and Innovation, 2019; Breidenbach et al., 2021). Consequently, not only does this result in AI research not involving people from key demographics who will be affected by these technologies, but the research performed on the generated language will lack important replicability for the evaluation process, where heterogeneous eval-

uation panels are not conducive to easy comparison across studies.[14] Whilst the substantial reliance on crowdsourcing platforms, particularly Mechanical Turk, may result in a different set of participants to the typical WEIRD demographic in social science and health research, the lack of reliable worker filters and allowances on what demographic data can be collected by requesters nevertheless results in a biased sample. Early works on the the use of crowdsourcing outside of this survey have even touted the low cost as an affordance of the method, whilst paying as low as 1/10th of a single US cent per individual HIT task (Callison-Burch, 2009). Perspectives such as these therefore result in biased demographics being used for evaluation and annotation at best, and the outright exploitation of people from particular socio-economic backgrounds and regions at worst.

In terms of inter-annotator agreement (IAA), we find that just over half of the surveyed papers include explicit mention of statistical tests for agreement levels. IAA is one method through which the variation amongst evaluator opinions can best be analysed, in order to understand how subjective a task is, and consequently how best to assess the quality of an NLG system's output where there is no "ground truth" for a given criteria (e.g. the humour, irony and sarcasm discussed here). Finally, we note that only 4 (of 17) papers convey any information regarding the level of training given to participants, whether that constitute the presence of qualifier tasks on crowdsourcing platforms (Sun et al., 2022a,b) or extensive documentation on the nature of the task (Oprea et al., 2021, 2022). In tasks that are inherently subjective, the provision of adequate training materials can help to account for variation in the understanding of different individuals by orientating opinions, and ensuring that all evaluators are working from the same definition of a particular phenomenon (such as distinguishing between general verbal irony and sarcasm specifically). Especially in cases where IAA is not reported, the inclusion of the extent of this background training given to evaluators is useful to the research community for judging to what extent the evaluators were in-line with each other's perspectives.

## 5 Discussion and Proposed Action Points

Our analysis has shown a distinct lack of reporting on potentially relevant demographic information in ad-

---

[13]This study paid 0.38 GBP for approximately 3 minutes of work. The USD conversion is the exchange rate at the time of writing this survey.

[14]We exclude the cases where the research was intended for a specific demographic group to start with, in which case a more homogeneous evaluator group may be desirable.

| Factor | # Papers | Percentage |
|--------|----------|------------|
| **Recruitment** | 12 | <71% |
| **Compensation** | 5 | <30% |
| **IAA** | 9 | <53% |
| **Training Materials** | 4 | <24% |

Table 5: Counts of papers containing explicit mention of recruitment methods, compensation details, inter-annotator agreement (IAA), and the provision of training materials.

dition to details concerning the evaluation procedure. To tackle this issue, we suggest future work on the evaluation of (especially subjective) generative tasks to include an "evaluation statement", akin to the data statements proposed by Bender and Friedman (2018). In line with their work, we suggest the inclusion of the following points for such statements:

- evaluation logistics

  - number of evaluators used
  - the split of samples across evaluators (e.g. does everyone rate every sample)
  - recruitment method used (e.g. internal emails, posters or crowdsourcing)
  - compensation given to evaluators (e.g. gift cards and cash - additionally, the choice to not monetarily compensate evaluators should also be divulged)

- evaluator demographic information

  - age (e.g. range, group or generation)
  - language proficiency & "native" language (L1)
  - gender
  - cultural background (e.g. religion, political leaning, country of origin and general education level)
  - task proficiency (i.e. expert or layperson)

- evaluation process

  - clear and operationalised definitions of the evaluated phenomena (with explicit mention if a term with a largely accepted definition is being used in a non-standard way)
  - amendments made in light of potential pilot studies
  - training materials and examples of criteria given to evaluators

- agreement study (i.e. reporting inter-annotator agreement)

While the language proficiency of the participants is relevant to all tasks, we recognise that other characteristics such as gender and cultural background (e.g. religion and political leaning) may not always apply. Still, it would be beneficial to report which facets were taken into consideration, as they may become relevant in hindsight, and in the case of dataset annotations, facilitate more extensive use of any collected data in future works. Similarly, task proficiency may sometimes seem irrelevant for subjective and intuitive tasks. However, Iskender et al. (2021) and Lloret et al. (2018) demonstrate the preference of expert evaluators for complex linguistic tasks. Additionally, authors may be reluctant to present only moderate agreement for subjective tasks due to concerns regarding evaluation quality. However, this criterion is highly relevant as it describes the expected degree of subjectivity for a task, a concept the community should be able to embrace. In addition to these points, we also encourage increased interest in the area of perspectivist approaches to the study of these language forms, and in NLP as a whole. Finally, the intentional omission of many of these criteria (such as IAA and language proficiency, in particular) may be considered intentionally deceptive, and therefore at odds with the foundations of open science.

## 6   Conclusion

In conclusion, we reiterate the importance of the role that Human Evaluation plays in assessing the outputs of Natural Language Generation systems. We have further demonstrated why particular types of language such as humour, irony, and sarcasm demand additional attention to be paid during the construction of human evaluation processes, particularly as it pertains to the demographic variables of selected evaluators. In no way do we discourage the use of the "non-native" or L2+ speakers of a target language in such evaluation (or any other minority demographic), nor do we likewise encourage the use of "native" or L1 target-language speakers (or any other majority demographic) as the true "gold standard". Rather, we aim to encourage additional thoughts and reflections in the community when selecting panels of human evaluators for such language, analysing results in light of their demographic characteristics as a feature, rather than a bug, whilst openly reporting such decisions wherever possible. To this end, we provide a list of suggested points that can be included as part of an "evaluation statement". Additionally, we advocate for the use of stratified sampling techniques to gain more varied evaluator panels.

## Limitations

The limitations of this work are as follows. We only survey papers from 2018-2023 in order to capture more recent and modern behaviours in the research discipline. However, this results in a small number of available papers. This is naturally exacerbated by the choice of focusing solely on humour, sarcasm and irony, rather than language generation more generally. However, we believe that surveys of NLG broadly have been performed in recent years to high standards, and instead wish to use these niche language types as specific examples of types of language that are immensely affected by these factors, whilst conveying that transparent reporting and consideration of evaluation practices is entirely applicable to NLG as a whole, and not just the subdomains we focus on. Furthermore, we focus exclusively on the use of human evaluators in generation tasks. However, much of the same ideas apply equally to the selection of annotators during dataset creation for detection tasks. All statements and opinions expressed in this work are by nature in good faith, and we accept, appreciate and welcome perspectives from other individuals from different backgrounds to support, challenge, or revise any of the claims made herein.

## Ethics Statement

We believe in and firmly adhere to the ACL Code of Conduct in the performance of this survey and the expression of opinions. We review only information that was publicly available in the published versions of the papers we discussed. However, whilst we encourage the collection of a lot of metadata about participants in NLG research, we emphasise that this data should only be collected where the correct ethics application procedures have been followed by the respective researchers, and that collected information should be only as fine-grained as is needed to accurately characterise the participant panel (for example, someone's hometown is not necessary when their nationality or cultural identity would suffice, and neither is someone's exact age when 10-year age brackets may be sufficient).

## Acknowledgements

Tyler Loakman is supported by the Centre for Doctoral Training in Speech and Language Technologies (SLT) and their Applications funded by UK Research and Innovation [grant number EP/S023062/1]. Aaron Maladry is supported by Ghent University [grant BOF.24Y.2021.0019.01.]

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

# A   Appendix

## A.1   Surveyed Generation Papers

| Title | Author (Year) |
|---|---|
| A Neural Approach to Pun Generation | Yu et al. (2018) |
| Punny Captions: Witty Wordplay in Image Descriptions | Chandrasekaran et al. (2018) |
| A Modular Architecture for Unsupervised Sarcasm Generation | Mishra et al. (2019) |
| Pun-GAN: Generative Adversarial Network for Pun Generation | Luo et al. (2019) |
| Pun Generation with Surprise | He et al. (2019) |
| Can Humor Prediction Datasets be used for Humor Generation? Humorous Headline Generation via Style Transfer | Weller et al. (2020) |
| R$^3$: Reverse, Retrieve, and Rank for Sarcasm Generation with Commonsense Knowledge | Chakrabarty et al. (2020) |
| Homophonic Pun Generation with Lexically Constrained Rewriting | Yu et al. (2020) |
| "Judge me by my size (noun), do you?" YodaLib: A Demographic-Aware Humor Generation Framework | Garimella et al. (2020) |
| Chandler: An Explainable Sarcastic Response Generator | Oprea et al. (2021) |
| Should a Chatbot be Sarcastic? Understanding User Preferences Towards Sarcasm Generation | Oprea et al. (2022) |
| Context-Situated Pun Generation | Sun et al. (2022b) |
| ExPUNations: Augmenting Puns with Keywords and Explanations | Sun et al. (2022a) |
| A Unified Framework for Pun Generation with Humor Principles | Tian et al. (2022) |
| AMBIPUN: Generating Puns with Ambiguous Context | Mittal et al. (2022) |
| "When Words Fail, Emojis Prevail": A Novel Architecture for Generating Sarcastic Sentences With Emoji Using Valence Reversal and Semantic Incongruity | Kader et al. (2023) |
| Multi-modal Sarcasm Generation: Dataset and Solution | Zhao et al. (2023b) |

Table 6: List of covered generation papers in the critical survey.