# OpenReview forum: "The Iron(ic) Melting Pot: Reviewing Human Evaluation in Humour, Irony and Sarcasm Generation"
_EMNLP/2023/Conference — EMNLP 2023 Findings_

### Official Review · Reviewer_bfpd · 2023-07-31

**Soundness:** 3

**Excitement:**

4: Strong: This paper deepens the understanding of some phenomenon or lowers the barriers to an existing research direction.

**Paper Topic And Main Contributions:**

The paper discusses the role of human evaluation in the assessment of humor, irony and sarcasm generation, leaving from the observation that while human evaluation is preferable to automated evaluation metrics, it is not infallible. The authors discuss how different evaluator demographics and user characteristics affect the interpretation of more esoteric forms of language, nevertheless the reporting of details which highly impact the transparency and replicability of user studies is often overlooked in research papers. The authors present a detailed description of humor, irony, and sarcasm, highlighting subtleties and explaining why they demand special attention in human evaluation studies. Then they survey papers from ACL Anthology published between 2018-2022, with a particular focus on 15 generation papers, with regards to the transparency of human evaluation in their reporting. The authors find that less than half of the papers (only 6 out of 15) report any demographic information, language proficiency, location, or evaluator level of education. Moreover, only 12 out of 15 papers report details on the recruitment and compensation of the annotators, while less than half include statistical tests for agreement levels. The authors conclude with the recommendation to pay special attention to the construction of human evaluation processes.


**Questions For The Authors:**

While your paper is particularly focused on the assessment of humor, irony and sarcasm, how do you think your recommendations translate to the design of user studies for other domains?


**Reasons To Accept:**

The paper is timely and addresses an important aspect (i.e. lack of transparency in the human evaluation process), it is well written and informative. The conference will certainly benefit from accepting this relevant piece of work. I highly enjoyed reading it!


**Reasons To Reject:**

While the discussion is very helpful and informative, it would be beneficial for the authors to include a list of concrete recommendations and takeaways for the reader on the design of future user studies that increase transparency in reporting assessments.

It would also be beneficial if the authors discuss how their findings could be informative for the design of user studies in other NLP tasks.

**Reproducibility:**

4: Could mostly reproduce the results, but there may be some variation because of sample variance or minor variations in their interpretation of the protocol or method.

**Reviewer Confidence:**

5: Positive that my evaluation is correct. I read the paper very carefully and I am very familiar with related work.

---

> ### Author Rebuttal · Authors · 2023-08-27
>
> We thank you for your review, and appreciate your kind words towards our work. We agree that formulating more concrete recommendations or “action points” would be a beneficial way to end the paper, and we intend to include such recommendations in the final version with the remaining space (see the end of our response).
>
> In terms of the effects on other NLG tasks, the need for transparency is universal, and increased information about evaluator panels would prove beneficial in any domain. Our selected language forms (such as humour) facilitate these claims to be made in the most obvious setting however. We additionally believe our main points translate over to other domains in terms of properly operationalising evaluation criteria (as other studies have looked at), as it is not always only the object of evaluation that is subject to personal interpretation, but often also what evaluators are specifically asked to rate language on (i.e. what is “funny” or “poetic” etc.). The design of specific NLG tasks in a way that avoids particular types of bias and interpretation differences is highly reliant on the exact task at hand, so it’s difficult to make blanket statements other than calling for transparency.
>
> Some of our suggested take-home points are as follows: Provide a short data evaluation statement, inspired by the data statements, such as the one proposed by Bender and Friedman (2018) for datasets annotated for detection tasks. Although a unified template for such statements could be useful, not all demographics are relevant for each task.
>
> Suggested Action Points:
> - Outline no. of evaluators, number of samples seen by each, and the method of recruiting them.
> - For each evaluator, describe demographics:
> age (range/group/generation)
> native language and proficiency in the target language
> Gender
> Cultural background (religion, political leaning, education level - The exact types should be chosen as necessary depending on the type of content being evaluated)
> Task proficiency/education -  For binary humour evaluation, one can assume a linguistic education does not have any significant benefits (due to being less representative of the general population) but this may be more relevant when evaluating specific types of humour that a linguist may be better placed to properly identify (for example, Iskender, Polzehl and Moller [2021, HumEval] and Lloret, Plaza and Aker [2018, LRE] demonstrate the preference of expert evaluators in more complex tasks).
>
> - Present the guidelines given to the evaluators
> - Definitions of the evaluated phenomena
> - Modifications made in light of potential pilot studies
> - Conduct a (at the very least minimal) agreement study to investigate the subjectivity of the evaluation task. Although presenting modest agreement may come across as limited evaluation quality, the community should be able to embrace the concept of subjectivity.
> Promote the use of perspectivist approaches that use the different demographic distributions of evaluator panels as a dimension of the research itself, rather than as a footnote detail. These systems are to eventually be used in the real world, and the real world is not homogenous, and nor should our NLP systems be expected to be.

---

### Official Review · Reviewer_Txig · 2023-08-04

**Soundness:** 4

**Excitement:**

4: Strong: This paper deepens the understanding of some phenomenon or lowers the barriers to an existing research direction.

**Paper Topic And Main Contributions:**

This position paper discusses the significance of human evaluation in Natural Language Generation (NLG) systems, particularly for humor, irony, and sarcasm. In the case of these "more esoteric forms of language", the authors perform a critical survey on recent works in NLG to assess how well evaluation procedures are reported in this subdomain, and stress the need for transparency by reporting evaluator demographics and note the overreliance on crowdsourcing for recruitment.


**Reasons To Accept:**

The paper is a position paper but falls very well within the scope of the conference. It is mostly a survey, but it justifies very well the motivation for the study, and results, even if restricted to the particular niche of NLG of humor/irony/sarcasm language, provide useful insight for the NLP community as a whole.

**Reasons To Reject:**

I have no reason to reject this paper.

**Reproducibility:**

5: Could easily reproduce the results.

**Reviewer Confidence:**

5: Positive that my evaluation is correct. I read the paper very carefully and I am very familiar with related work.

**Typos Grammar Style And Presentation Improvements:**

There are some minor aspects that could be clarified and improve the paper's already remarkable clarity and insightfulness:

(l. = line)
l.105: "demand characteristics in evaluators": unclear formulation (would you elaborate?)

l.120-3: "resulting in biased perspectives when evaluating text quality that is only somewhat *overcome* with the provision of training materials." : unclear formulation

l.333-5: "lower power distance (i.e. social hierarchy)... higher power distance": unclear formulation, particularly, without any qualification social hierarchy.

---

> ### Author Rebuttal · Authors · 2023-08-27
>
> We thank you for your review and highly appreciate your kind comments towards our work. Thank you for identifying clarity issues, particularly where we have adopted terms from the social sciences and not made sure to define them. To quickly summarise:
>
> L105) “demand characteristics” are clues in an experiment that may accidentally inform a participant of the “real” aim of a study and desirable results, therefore making the participant alter their behaviour so that it falls in line with what they believe the research is trying to find. This may effectively lead to research coming to whatever conclusions the researchers wanted to reach, because the evaluators are not necessarily answering objectively. This is particularly true for crowdsourcing platforms, where workers (whether correctly or not) assume that they are more likely to be chosen for future tasks if they give researchers “better” results.
>
> L120) Here we mean that training materials/guidelines can help to converge participants on a common definition of a certain concept (e.g. sarcasm), but this cannot entirely overpower individual differences, and are therefore not a fool proof solution.
>
> L333) “Power Distance” refers to the distribution of “power” (political, financial etc.) across a society. A lower power distance means that members of a society are more “equal”. Take, for example, a middle class person in a democratic nation such as the US. The power distance between this member of the public and a local politician is likely to be lower than the same social distance would be in a nation run by an authoritarian dictatorship (on average, anyway). We hope to clarify and reformulate these points in the final version.

---

### Official Review · Reviewer_4dTM · 2023-08-10

**Soundness:** 2

**Excitement:**

2: Mediocre: This paper makes marginal contributions (vs non-contemporaneous work), so I would rather not see it in the conference.

**Missing References:**

Meaney, J. A. (2020, July). Crossing the Line: Where do Demographic Variables Fit into Humor Detection?. In Proceedings of the 58th Annual Meeting of the Association for Computational Linguistics: Student Research Workshop (pp. 176-181).

Shani, C., Libov, A., Tolmach, S., Lewin-Eytan, L., Maarek, Y., & Shahaf, D. (2022, April). “Alexa, do you want to build a snowman?” Characterizing playful requests to conversational agents. In CHI Conference on Human Factors in Computing Systems Extended Abstracts (pp. 1-7).

Shapira, N., Kalinsky, O., Libov, A., Shani, C., & Tolmach, S. (2023, March). Evaluating humorous response generation to playful shopping requests. In European Conference on Information Retrieval (pp. 617-626). Cham: Springer Nature Switzerland.

**Paper Topic And Main Contributions:**

The Iron(ic)Melting Pot: Reviewing Human Evaluation in Humour, Irony and Sarcasm Generation

This position paper claims the importance of reporting demographics when using human evaluation for NLG tasks in the domain of humor, irony, and sarcasm.
The authors presented an overview of the topics (humor, irony, sarcasm) and gave 1-5 examples for each.
The authors review the literature (critical survey), showing that most papers do not include demographics.

**Reasons To Accept:**

There is a summary of the topics of humor, irony, and sarcasm.
I believe that the claim (that it is important to refer to demographics) is correct and it is important to raise awareness.

**Reasons To Reject:**

The importance of demographic variables within humor is known (e.g., meaney 2020). The survey is based on only 15 papers and is very unrepresentative. The NLP computational humor community is larger than was taken into consideration. For example NLP humor detection in HCI (Shani et al.,  2022) or NLP humor generation in ECIR (Shapira et al., 2023). There are much more venues to take into account.

There is room for improvement in validating the position with more effective efforts. For example, the inclusion of experiments showcasing the impact of demographics. For instance, employing humor-related datasets and having diverse groups assess them could illustrate varying outcomes.

---

After reading the author's response, my main criticism remains.
I would also want to respond regarding the authors' claim _"provide evidence that current evaluation is frequently sub-optimal"_

Pointing to a study that did not report demographics
1. Does not mean that the study did not take into account the matter of demographics
2. Does not prove the importance of the report

The authors did not provide _any_ empirical experiment regarding the importance of demographics. The anecdotal examples the authors provided not convincing enough in my opinion.

The importance of demographic variables within humor is known (e.g., meaney 2020). The survey is based on only 15 papers and is very unrepresentative. The NLP computational humor community is larger than was taken into consideration. For example NLP humor detection in HCI (Shani et al.,  2022) or NLP humor generation in ECIR (Shapira et al., 2023). There are much more venues to take into account.

There is room for improvement in validating the position with more effective efforts. For example, the inclusion of experiments showcasing the impact of demographics. For instance, employing humor-related datasets and having diverse groups assess them could illustrate varying outcomes.

**Reproducibility:**

4: Could mostly reproduce the results, but there may be some variation because of sample variance or minor variations in their interpretation of the protocol or method.

**Reviewer Confidence:**

4: Quite sure. I tried to check the important points carefully. It's unlikely, though conceivable, that I missed something that should affect my ratings.

**Typos Grammar Style And Presentation Improvements:**

Regarding the examples in Tables 1-3: It would be better if you could emphasize examples that are really impossible to understand without a demographic context. Personally, I was able to understand the humor/sarcasm in all provided 12 examples. It could have been much more powerful and convincing if there had been examples that I would not have understood without explanation. Maybe even divide it by different demographics - which examples will get different scores in different demographics and also prove it with an experiment.

---

> ### Author Rebuttal · Authors · 2023-08-27
>
> Thank you for your critical review. Firstly, we accept that the community is indeed larger than what was surveyed, However, in line with standard methodology for performing critical reviews (Moher et al., 2009 - The PRISMA Statement), we defined our scope before performing the review, in which we selected a subset of the ACL Anthology which included all *ACL venues, EMNLP, LREC and COLING. As we state, this resulted in 244 papers across main conference, findings, system demos, and workshops, which was filtered down to 15 as we deemed detection papers to be out of scope, in addition to removing shared task submissions due to using a unified evaluation strategy, and finally, we remove 4 generation papers due to not clearly using human evaluation. It is also not unusual for small sample sizes to be used when supporting a point about the inadequacy of an approach, for example, Reiter (2018, in Computational Linguistics) use 34 papers to analyse the use of BLEU, which is only a subset of all papers that would have been reporting BLEU (as is acknowledged in their paper). We also present our work primarily as a position/opinion paper that identifies a problem and offers justification via the small survey, rather than being primarily a survey paper that is intended to represent the entire domain.
>
> We intentionally focus on generation instead of detection because the human evaluation for detection tasks is more static. By static, we mean that the human evaluation is unified for each dataset and remains the same for any system that is evaluated on that specific dataset (ideally, the publication of a dataset also includes a data statement, such as the one proposed by Bender and Friedman, 2018). This makes it somewhat easier and more reliable to compare system performance. For generation, each system generates its own samples and is evaluated by different annotators, likely with different demographics.
>
> Our aim was to provide evidence that current evaluation is frequently sub-optimal, and the fact we were able to find so many significant oversights in the small remaining sample from top-tier venues we believe strengthens our main points, rather than weakens them. We of course agree that a wider catchment area would better represent the state of the research domain in its entirety, but we do not feel as though this undermines the essential points we have tried to convey about the special consideration that esoteric language should be given at evaluation time.
>
> Additionally, we agree that experimentation to demonstrate the effects of these variables empirically would be valuable, but regretfully admit that time and resource constraints made this untenable due to the large participant pool that would be required (as having single individuals represent an entire demographic is problematic, and many characteristics may co-occur).
>
> Finally, our examples were designed to be accessible by the majority of the audience so that it is clear how variables that are often taken for granted actually cause differences in interpretations. As we state, simply changing elements of our examples such as through using government officials from less widely known nations than the U.S would immediately present an example of humour where the meaning would be lost on many individuals. We will aim to include a few examples of subculture specific language in the final version, particularly for humour (as sarcasm and irony often require more context specific information, rather than world knowledge, and therefore the difference between a genuine compliment and a sarcastic retort can simply be the person uttering it).
>
> We also thank you for the recommended reading, and look to incorporate these into the final version where appropriate (especially in regards to Meaney, 2020).

---

### Meta-Review · Area_Chair_ztqn · 2023-09-19

**Recommendation:** 4

**Metareview:**

This is a position paper arguing that characteristics of evaluators (i.e., demographic characteristics) are especially important for generation of language such as humour, irony, and sarcasm.

Overall, this paper makes a valuable contribution. Reviewers agree that the paper’s core claim is well-motivated, and that it addresses a critical issue (lack of transparency with regards to human evaluators) and raises important awareness on the issue. Reviewers’ concerns are generally well-addressed by the authors’ responses; one reviewer suggests that the work would be strengthened by a list of concrete takeaways, which the authors provide in their response, while another reviewer’s concern about the scope of the literature review is addressed by an extended response detailing the review methodology.

Finally, one reviewer raises concerns that the position paper requires an empirical experiment in order to justify its claim (i.e., that evaluators’ demographic characteristics impact outcomes), on the grounds that the paper’s analysis — which provides examples showing that different demographic characteristics impact interpretations of humour, irony, and sarcasm, and further shows that most papers in the meta-analysis do not mention demographic characteristics — is not convincing. While I agree that such an empirical experiment might be interesting, I see it it as a different analysis (for another paper, perhaps) and not one that is required for this paper. While it’s perhaps true that a study that does not report demographics (as is the case for most of the studies examined by this paper) may still have taken demographics into account or may have deemed demographics irrelevant to its analysis, the burden is on such a study to report this thinking, and not on readers to assume that it was present — therefore, in my view this paper’s analysis showing a distinct lack of reporting in the literature, as well as examples illustrating that demographics are indeed often relevant, seems to justify its claim.

---

### Decision · Program_Chairs · 2023-10-07

**Decision:**

Accept-Findings

**Comment:**

This is a position paper arguing that characteristics of evaluators (i.e., demographic characteristics) are especially important for generation of language such as humour, irony, and sarcasm.

Overall, this paper makes a valuable contribution. Reviewers agree that the paper’s core claim is well-motivated, and that it addresses a critical issue (lack of transparency with regards to human evaluators) and raises important awareness on the issue. Reviewers’ concerns are generally well-addressed by the authors’ responses; one reviewer suggests that the work would be strengthened by a list of concrete takeaways, which the authors provide in their response, while another reviewer’s concern about the scope of the literature review is addressed by an extended response detailing the review methodology.

Finally, one reviewer raises concerns that the position paper requires an empirical experiment in order to justify its claim (i.e., that evaluators’ demographic characteristics impact outcomes), on the grounds that the paper’s analysis — which provides examples showing that different demographic characteristics impact interpretations of humour, irony, and sarcasm, and further shows that most papers in the meta-analysis do not mention demographic characteristics — is not convincing. While I agree that such an empirical experiment might be interesting, I see it it as a different analysis (for another paper, perhaps) and not one that is required for this paper. While it’s perhaps true that a study that does not report demographics (as is the case for most of the studies examined by this paper) may still have taken demographics into account or may have deemed demographics irrelevant to its analysis, the burden is on such a study to report this thinking, and not on readers to assume that it was present — therefore, in my view this paper’s analysis showing a distinct lack of reporting in the literature, as well as examples illustrating that demographics are indeed often relevant, seems to justify its claim.